# An integrated continuity of care measure improves performance in models predicting medication adherence using population-based administrative data

**Shenzhen Yao**[1☯], **Lisa Lix**[2‡], **Gary Teare**[3‡], **Charity Evans**[1‡], **David Blackburn**[1☯]*

**1** College of Pharmacy and Nutrition, University of Saskatchewan, Saskatoon, Saskatchewan, Canada, **2** Department of Community Health Sciences, Rady Faculty of Health Sciences, University of Manitoba, Winnipeg, Manitoba, Canada, **3** Program Knowledge, Evidence and Innovation, Provincial Population and Public Health, Alberta Health Services, Calgary, Alberta, Canada

☯ These authors contributed equally to this work.
‡ LL, GT and CE also contributed equally to this work.
* d.blackburn@usask.ca

**Data Availability Statement:** Data cannot be shared publicly because of protection of personal information by the Protection of Privacy Act of Saskatchewan. Data are available at the

## Abstract

### Objectives

Continuity of care (COC) is considered an important determinant of medication adherence based on measures such as the usual provider continuity index (UPCI) that are derived exclusively from physician visit claims. This study aimed to: a) determine if high UPCI values predict physicians who deliver different clinical services; and b) compare UPCI with an integrated COC measure capturing physician visits, prescribing, and a complete medical examination in a multivariable model of patients receiving statin medications.

### Methods

This was a retrospective cohort study of new statin users between 2012 and 2017 in Saskatchewan, Canada. We calculated sensitivity/specificity of a high UPCI value for predicting physicians who were prescribers of statins and/or providers of complete medical examinations. Next, we used logistic regression models to test two measures of COC (high UPCI value or an integrated COC measure) on the outcome of optimal statin adherence (proportion of days covered ≥80%). The DeLong test was used to compare predictive performance of the two models.

### Results

Among 55,144 new statin users, a high UPCI was neither a sensitive or specific marker of physicians who prescribed statins or performed a complete medical examination. The integrated COC measure had a stronger association with optimal adherence [adjusted odds ratio (OR) = 1.56, 95% confidence interval (CI) 1.50 to 1.63] than UPCI (adjusted OR = 1.23, 95% CI 1.19 to 1.28), and improved predictive performance of the adherence model.

Saskatchewan Health Quality Council (contact via info@hqc.sk.ca) under data sharing agreements with the Saskatchewan Ministry of Health and eHealth Saskatchewan for researchers who meet the criteria for access to confidential data.

**Funding:** This study was funded by the University of Saskatchewan Chair in Patient Adherence to Drug Therapy to DB.

**Competing interests:** I have read the journal's policy and the authors of this manuscript have the following competing interests: David Blackburn is the Chair in Patient Adherence to Drug Therapy within the College of Pharmacy and Nutrition, University of Saskatchewan. This position was created through unrestricted financial support from AstraZeneca Canada, Merck Canada, Pfizer Canada, and the Province of Saskatchewan's Ministry of Health. None of the sponsors were involved in developing this study or writing the manuscript. Shenzhen Yao, Lisa Lix, Gary Teare, and Charity Evans declare no conflicts. This does not alter our adherence to PLOS ONE policies on sharing data and materials. Our study data contain sensitive patient information and is only available upon approval. For data request please contact Saskatchewan Health Quality Council at info@hqc.sk.ca.

## Conclusion

The number of physician visits alone appears to be insufficient to represent COC. An integrated measure improves predictive performance for optimal medication adherence in patients initiating statins.

## Introduction

Non-adherence is defined as the failure to take medications according to the prescribed regimen [1]. It occurs in up to half of all people with chronic conditions and is responsible for $100 to 500 billion in avoidable healthcare costs annually in the US [2]. Patients exhibiting non-adherence experience higher rates of hospitalization, death, and higher healthcare costs [3–5]. Despite a strong theoretical framework, understanding how healthcare practices precisely influence adherence remains a challenge.

Studies suggest that individual physicians can improve medication adherence by establishing continuity of care (COC) for their patients [6–9]. The precise nature of this association is unknown but is likely mediated by factors promoting a strong relationship between patients and physicians [10, 11]. Indeed, an ongoing relationship between a physician and a patient is associated with higher satisfaction, improved trust, and more effective communication [12]. Having a single physician also helps ensure the completeness of a patient's health records, and can facilitate the coordination of disease management activities [13].

COC is a complex concept [14]. Although previous studies have demonstrated a positive correlation with medication adherence, conventional measures of COC have limitations that may be improved upon using a more comprehensive definition that is specific for medication adherence [6–9]. COC is commonly measured by the usual provider continuity index (UPCI) [14, 15]. UPCI is determined by a simple calculation of the percentage of visits to a specific physician relative to all other physician visits in a given time period [14, 15]. As a result, it is highly influenced by total number of visits, and total number of different physicians [15]. For example, a patient with multiple chronic conditions may visit many different physicians in a given year. In this situation, the UPCI for that patient's regular physician may be low because the denominator (i.e., total number of visits with all physicians) is increased compared to a different patient who visits a single physician exclusively. Moreover, since the UPCI is based solely on visit occurrences, the nature of the visits is not accounted for. The UPCI approach does not consider prescribing activities despite evidence suggesting that individuals are more likely to be adherent if their regular physician is the prescriber of their treatment regimes [16]. Certainly, it seems logical that a continuity of care measure applied to a cohort of medication users should consider the physician's prescribing activities relating to the drug(s) of interest. Further, the UPCI does not represent clinical services such as complete medical examinations (CME), which would be expected from a patient's regular physician. Although this activity has been identified as a measure of COC, few studies have examined the impact of CME providers on medication adherence [17, 18]. Although previous studies have attempted to improve on measures of COC with minimal success, the updated definitions have continued to focus on visit frequency only [6, 9].

We hypothesized that 1) a high UPCI value will perform poorly in predicting physicians who provide other clinical activities to specific patients (i.e., prescribing, and complete medical examinations); and 2) an integrated COC measure consisting of physician visits, prescribing, and claims for a complete medical examination would result in a stronger association with

medication adherence, and improve the predictive ability of medication adherence models. The objectives of this study were: 1) to examine the accuracy of UPCI for predicting other COC-related clinical activities, including prescribing statin medications, and/or performing complete medical examinations; and, 2) to determine if an integrated measure of COC is superior to UPCI in discriminating adherent statin users, and therefore improving the predictive performance of a covariate-adjusted model of adherence to statins.

## Methods

### Data sources

Study data were extracted from administrative databases for the province of Saskatchewan, Canada. These databases include the person registration file, the physician service claims file, the hospital discharge abstract database, the emergency service file and the prescription drug claims files [19].

**Person registration file.** The person registration file captures birth, sex, rural/urban residence, health insurance coverage start/end dates, and median household income quintiles estimated by linking the first three digits of postal code to Statistics Canada Census data [20]. Although area-based estimation of income is less accurate than direct household tax records [21], this was the only approach available in our population-based databases to attempt to control for the important effect of socioeconomic status [22].

**Physician service claims file.** The physician service claims file captures the date of the service, the type of the service (in-hospital or out-patient), the diagnosis of the service using three-digit, 9$^{th}$ version of International Classification of Diseases (ICD-9) codes [23], the encrypted identification code of the service provider, the specialty of the service provider, the fee code for billing, and the type of payment to the provider.

**Hospital discharge abstract database.** The hospital discharge abstract database captures admission and discharge date, up to 25 diagnoses by ICD-9 or 10th Canadian version of International Statistical Classification of Diseases (ICD-10-CA) codes [23, 24], and an indicator on whether the recorded event was for acute care or alternative care (i.e., a patient was occupying a bed in a hospital and did not require the intensity of services as for acute care) [25].

**Emergency service file.** The emergency service file captures admission and discharge date of visits to emergency departments. It also contains a field for main responsible diagnosis of the visit in ICD-10-CA codes.

**Prescription drug claims files.** The prescription drug claims files capture dispensations of prescription medications in out-patient settings. Each claim includes a Health Canada drug identification number (DIN), a dispensation date, the quantity dispensed, total cost (including medication acquisition cost and markup/dispensing fee), the encrypted identification code of the prescriber (the same physician as in the physician service claims file), and the proportion covered by government insurance.

### Study design and population

A retrospective cohort was conducted consisting of individuals who initiated a new 3-hydroxy-3-methylglutaryl-coenzyme (HMG CoA) reductase inhibitor medication (i.e., statin) between January 1, 2012, and December 31, 2017. A new user was defined as receiving no dispensations for a statin medication in the five years prior. Statins were used as the non-adherence model for several reasons: they are typically indicated for lifelong treatment, the dosing strategy does not change according to symptoms, no therapeutic equivalent existed during the period of study, they are administered once per day, they are prescribed to a large percentage of the adult population regardless of gender or age, and they have been the target of

extensive research in the field of medication adherence. Finally, statins are associated with reduced morbidity and mortality from atherosclerotic cardiovascular disease so adherence is important and relevant to population health [26–29].

The date of the earliest dispensation of a statin medication was the index date, and patients were followed for 365 days. The cohort exclusion criteria were: 1) missing age or sex information in the person registration file; 2) age on the index date less than 18 years; 3) not continuously registered in the provincial health plan during five years prior to the index date, or the one-year follow-up period; 4) admitted to a long term care facility within five years prior to the index date, or 365 day follow-up period; 5) admitted to an out-province hospital during the 365 day follow-up; 6) a claim for pregnancy (ICD-9: 641–676, V27; ICD-10 and ICD-10-CA: O1, O21-95, O98, O99, Z37) in the 365 days prior to the index date or in the 365 days after the index date [30]; or 7) no visits to a general practitioner (GP) during the 365 day follow-up period.

## COC measures and physician classifications

For each patient, we defined the following COC measures: a) usual care provider and the UPCI [14, 15]; b) usual statin prescriber (USP); c) complete medical examination provider (CMEP) [17]; and d) an integrated COC measure that combined all three measures (i.e., a single GP identified as the usual care provider, USP, and CMEP).

For determination of the usual care provider, we first identified all distinct service claims provided by GP physicians during each patient's follow-up period. Multiple claims by the same GP for the same patient on the same date were treated as one visit [15]. Service claims were not included if: 1) the claim was marked as invalid in the database; 2) if the service was provided to a hospitalized patient; or 3) if the claim originated from an out-of-province provider. For each patient, a usual care provider was identified as the GP with the most frequent visits during the follow-up period. In the case of a tie, multiple GPs could be assigned as usual care providers for a given patient.

Next, a UPCI value was calculated for each patient by the following formula: $UPCI = \frac{n_{max}}{N}$ [15]; where $n_{max}$ was the number of visits between the patient and the most frequently visited GP (i.e., the usual care provider) within the follow-up period and N was the total number of visits between the patient and all GP physicians visited within the same period. Based on the calculated UPCI value, each patient was assigned into a high or low UPCI category using the median UPCI value of the study cohort as the cut off. This process has been used previously to measure COC [15].

The usual statin prescriber (USP) was any type of physician (i.e., not necessarily a GP) of a patient listed on the highest number of statin dispensation claims during the follow-up period. In cases where a tie was observed, more than one physician was identified as USPs. Complete medical examination providers (CMEP) were identified on at least one claim for a complete medical examination during the follow-up period (i.e., a fee code billed for complete assessment, or chronic disease management) [31]. A patient could have multiple CMEPs within the study period [17], and any type of physician listed in the physician service claims was considered (i.e., not necessarily a GP). Finally, we combined these definitions into an integrated COC measure (yes/no) depending on whether a single GP was identified as: 1) the usual care provider; 2) the USP; and 3) the CMEP [17].

## Outcome measures

The study outcome was optimal adherence to statin medications defined as proportion of days covered (PDC) > = 80% [32, 33]. PDC was calculated for the 365-day period from the index

date for each patient. As these drugs are typically prescribed once daily, the number of days supplied during this time was estimated from the total quantity of tablets dispensed [34]. Quantities dispensed near the end of the follow-up period were truncated based on the number of days remaining in the follow-up period. Switching between statin medications was allowed. The total number of statin tablets dispensed was divided by 365 days (minus days spent in hospital) to obtain the adherence percentage. Details of the PDC method have been described, and validated previously [32, 33].

## Covariates

We built a multivariable model with covariates previously used to predict medication adherence from administrative databases [35]. These covariates were organized under a framework with five categories: patient, socioeconomic status, treatment, healthcare system, and condition factors [35]. The covariates were measured in the period up to 365 days prior to the index date if not otherwise specified. The patient covariates were age, sex, and residence (rural/urban) on the index date. The socioeconomic status covariates were income level, which was based on neighborhood median household income quintiles (lowest = 1, highest = 5) on the index date [36, 37]. The treatment covariates were number of distinct prescription medications, which were determined from unique drug identification numbers. The healthcare system covariates were number of out-patient visits (to GPs and to specialists, respectively), and percentage of prescription medication cost paid by government health insurance. The condition covariates were number of hospitalizations for acute care, number of emergency department visits, Charlson comorbidity score, and clinical conditions (yes/no) identified from published models of medication adherence [38]. These clinical conditions were osteoporosis, rheumatoid arthritis, hypertension, stroke, ischemic heart disease, acute myocardial infarction, heart failure, multiple sclerosis, Parkinson's disease, Alzheimer's disease and dementia, epilepsy, asthma, chronic obstructive pulmonary disease, diabetes, mood and anxiety diseases, schizophrenia, and cancer [30]. These clinical conditions were identified using validated case definitions provided by the Canadian Chronic Disease Surveillance System and were based on diagnoses recorded in the service claims file and hospital discharge abstract database, and medications in the prescription drug claims dating back to January 1st, 1996 [30].

## Statistical analysis

We described the baseline characteristics of the study cohort using descriptive statistics for all patients as well as subgroups based on COC measured by UPCI and the integrated COC measure. These characteristics included median age, percentages by sex (female/male), residence (rural/urban), and median income quintile (1 = lowest, 5 = highest). We also described the use of health services, including the percentage of patients with one or more hospitalizations for acute care (0, or $> = 1$), the percentage with one or more visits to emergency department, the median number of visits to GPs, and the median number of visits to specialists.

To determine if a high UPCI value was predictive of patients receiving various clinical activities from a given physician, we calculated its sensitivity, specificity, positive predictive value (PPV), negative predictive value (NPV), and the Kappa statistic [39] for predicting the usual statin prescriber (USP), the CMEP, and the integrated COC measure as the reference standards.

Next, we built logistic regression models to test the effect by two measures of COC on optimal adherence to statins: a high UPCI value (representing traditional measures of COC) versus presence of the integrated COC measure. Both unadjusted and adjusted models were tested separately for each COC measure as the independent variable. The unadjusted models had a

single COC measure as the explanatory variable. The adjusted model had the COC measure plus all covariates described above. To maximize the control for confounding, all covariates were included regardless of statistical significance in each of the two adjusted models, except for those exhibiting collinearity with the independent variable. Multicollinearity between COC and the adherence covariates were examined by the variance inflation factor (VIF) obtained from a regression model. If the VIF value was greater than 2.5, the covariate was removed [40]. Odds ratios (ORs) and 95% confidence intervals (95% CIs) are reported.

We performed the DeLong test to compare predictive performance of the two adjusted models that each contained a COC measure [41]. The model that produced a larger estimate of the area under the receiver operating characteristic curve (AUROC) was considered to have better predictive performance if the difference in the AUC estimates was statistically significant (p<0.05) [41].

To test the consistency of our results, several subgroup and sensitivity analyses were conducted. In subgroup analyses, we assessed the impact of an integrated COC measure (yes/no) among patients with a high UPCI and with a low UPCI separately. We also assessed the impact of UPCI (high/low) among patients with and without integrated COC. We conducted sensitivity analyses for a modified adherence measure, which was recalculated without allowing accumulated supplies between refills, and changed the threshold for "low" UPCI level to the 25th and 75th percentile rather than the median. SAS statistical software, version 9.4, (SAS Institute Inc., Cary, NC, USA) was used to conduct all analyses [42].

## Ethical considerations

Ethics approval (certificate number: 14–143) was granted by the University of Saskatchewan Biomedical Research Ethics Board (REB). Data access was granted at the Saskatchewan Health Quality Council under data sharing agreements with the Saskatchewan Ministry of Health and eHealth Saskatchewan. The University of Saskatchewan REB approved to waive the informed consent from study participants for the following reasons: 1) It was a retrospective study using historical data dated back to 1996; 2) All participants were anonymized by encrypted IDs; 3) Privacy of individuals was further protected by suppressing results from any group of fewer than six participants.

## Results

Overall 180,010 patients received statin medications between January 1, 2012, and December 31, 2017. Among them, 21,149 (11.8%) were excluded due to missing demographic information, death during the study period, or a lack of continuous beneficiary status (Fig 1). The final cohort was comprised of 55,144 (30.6% of 180,010) new users of statin medications. The median age of the final cohort at the index date was 59.0 years [interquartile range (IQR) 51.0 to 67.0], 44.2% (24,385/55,144) were females and 32.3% (17,811/55,144) lived in a rural setting. The median number of GP visits in the 365 day period prior to index date was 6.0 (IQR 3.0 to 9.0, Table 1).

A single usual care provider (i.e., the GP with the highest number of visits) was identified for 92.6% (n = 51,071) of the cohort, whereas 7.4% (n = 4,073) of patients had two or more GPs tied for the highest number of visits. The median UPCI among the cohort was 0.82 (IQR 0.62 to 1.00), meaning half of all patients visited the same physician for 82% to 100% of their total GP visits during the one-year follow-up period. Similarly, a single usual statin prescriber (USP) could be identified for the vast majority of patients (n = 52,693, 95.6%). In contrast, only 22,017 (39.9%) of the patients received complete medical examinations from a GP physician. The rest 33,127 (60.1%) of the patients either had no complete medical examinations

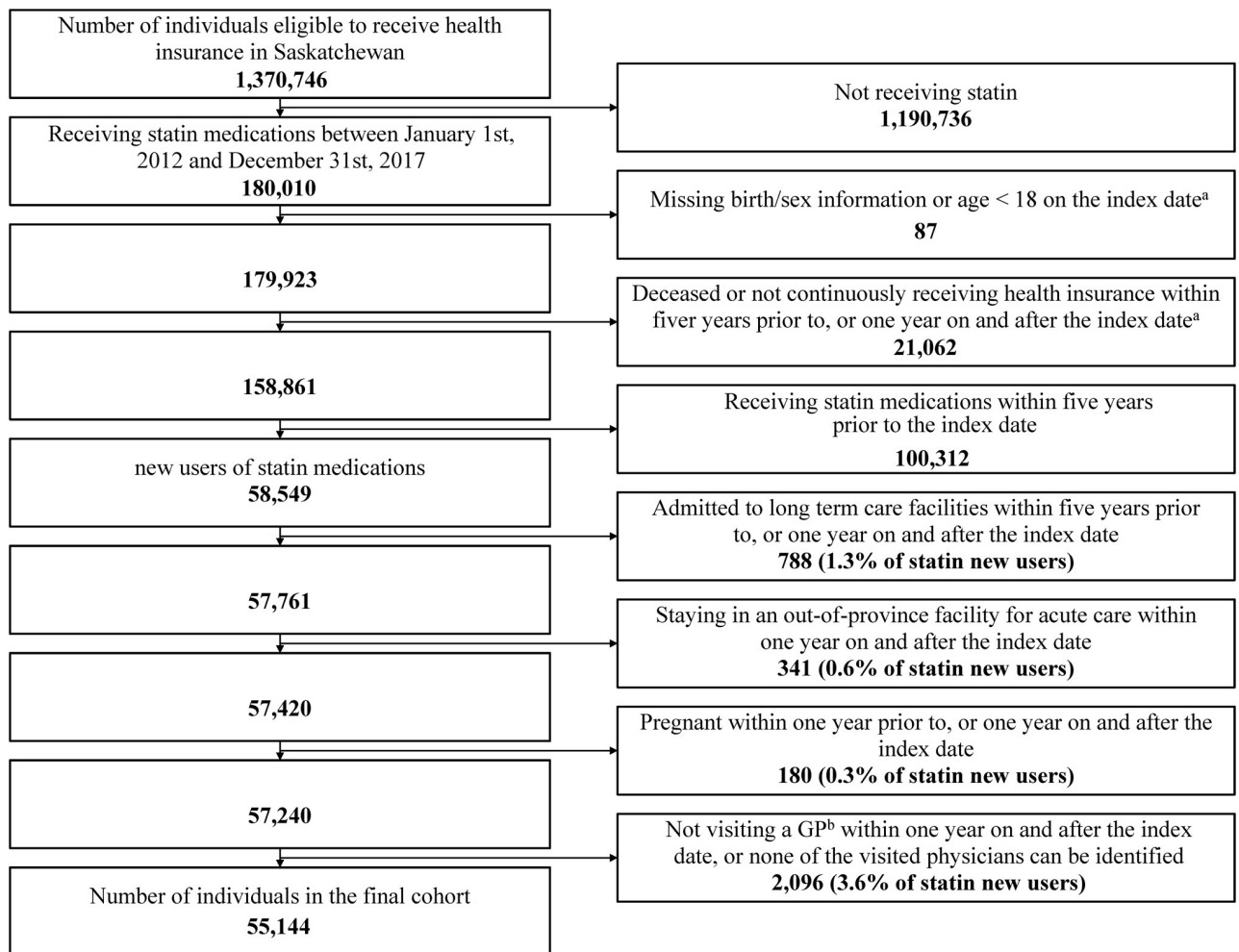

**Fig 1. Cohort flow chart.** [a]Index date = the earliest date receiving a statin medication between January 1st, 2012 and December 31st, 2017;
[b]GP = general practitioner.

during the follow-up period (322,41, 58.5% of 55,144) or received the examinations from a specialist (886, 1.6% of 55,144). Finally, 15,579 (28.3%) of the patients were classified as receiving integrated COC, defined as having a single GP for their usual care provider, USP, and CMEP.

A high UPCI (i.e., above the median value) was neither a sensitive or specific marker to identify a physician who was also the USP or CMEP [Table 2]. The sensitivity ranged from 0.55 (95% CI 0.55 to 0.56, using UPCI to predict usual statin provider) to 0.58 (95% CI 0.58 to 0.59, using UPCI to predict integrated COC). The specificity ranged from 0.52 (95% CI 0.51 to 0.52, using UPCI to predict CMEP) to 0.61 (95% CI 0.60 to 0.62, using UPCI to predict the usual statin provider, Table 2).

Both high UPCI and the integrated COC measure showed statistically significant associations with optimal adherence to statin medications. Optimal adherence was observed in 56.0% (15,606/27,859) of patients with a high UPCI versus 49.9% (13,604/27,285) of those with low UPCI (unadjusted OR = 1.28, 95% CI 1.24 to 1.32, adjusted OR = 1.23, 95% CI 1.19 to 1.28). In comparison, a stronger association with optimal adherence was observed when UPCI was included in the integrated COC measure (unadjusted OR = 1.45, 95% CI 1.40 to 1.51, adjusted

**Table 1. Baseline characteristics[a] of the final cohort.**

| | All | Patients grouped by UPCI[b] | | Patients grouped by integrated COC[c] | |
|---|---|---|---|---|---|
| | | High(> = 0.82) | Low(<0.82) | Yes | No |
| | n = 55,144 | n = 27,859 | n = 27,285 | n = 15,579 | n = 39,565 |
| Median age (IQR[d]) | 59.0 (51.0, 67.0) | 59.0 (52.0, 68.0) | 58.0 (50.0, 67.0) | 59.0 (51.0, 67.0) | 59.0 (51.0, 67.0) |
| Females (n, %) | 24,385 (44.2) | 11,635 (41.8) | 12,750 (46.7) | 6,840 (43.9) | 17,545 (44.3) |
| Patients with one or more hospitalizations for acute care (n, %) | 12,528 (22.7) | 6,203 (22.3) | 6,325 (23.2) | 2,626 (16.9) | 9,902 (25.0) |
| Visits to GPs[e], median (IQR) | 6.0 (3.0, 9.0) | 5.0 (3.0, 9.0) | 6.0 (3.0, 10.0) | 6.0 (3.0, 9.0) | 5.0 (3.0, 9.0) |
| Visits to specialists, Median (IQR) | 2.0 (0.0, 6.0) | 2.0 (0.0, 6.0) | 2.0 (0.0, 6.0) | 2.0 (0.0, 5.0) | 2.0 (0.0, 7.0) |
| Patients with one or more visits to emergency department (n, %) | 11,450 (20.8) | 5,519 (19.8) | 5,931 (21.7) | 2,739 (17.6) | 8,711 (22.0) |
| Patients by income level (n, %) | | | | | |
| 1 (lowest) | 10,339 (18.7) | 4,787 (17.2) | 5,552 (20.3) | 2,675 (17.2) | 7,664 (19.4) |
| 2 | 10,207 (18.5) | 5,058 (18.2) | 5,149 (18.9) | 2,761 (17.7) | 7,446 (18.8) |
| 3 | 10,093 (18.3) | 5,182 (18.6) | 4,911 (18.0) | 2,942 (18.9) | 7,151 (18.1) |
| 4 | 11,289 (20.5) | 5,897 (21.2) | 5,392 (19.8) | 3,251 (20.9) | 8,038 (20.3) |
| 5 (highest) | 10,268 (18.6) | 5,456 (19.6) | 4,812 (17.6) | 3,052 (19.6) | 7,216 (18.2) |
| missing | 2,948 (5.3) | 1,479 (5.3) | 1,469 (5.4) | 898 (5.8) | 2,050 (5.2) |
| Patients by residence location (n, %) | | | | | |
| Rural | 17,811 (32.3) | 8,666 (31.1) | 9,145 (33.5) | 4,364 (28.0) | 13,447 (34.0) |
| Urban | 37,333 (67.7) | 19,193 (68.9) | 18,140 (66.5) | 11,215 (72.0) | 26,118 (66.0) |

[a] Median age, number of females, residence (rural/urban), and patient income level were measured on the index date; Number of patients with one or more hospitalizations, median visits to GPs/specialists, patients with one or more visits to emergency departments were measured within one year prior to the index date;

[b] UPCI = usual provider continuity index;

[c] COC = continuity of care;

[d] IQR = interquartile range;

[e] GP = general practitioners.

OR = 1.56, 95% CI 1.50 to 1.63). Optimal adherence was observed in 59.5% (9,277/15,579) of patients meeting the integrated COC criteria versus 50.4% (19,933/39,565) of those who did not.

The significant association between the integrated measure of COC and optimal adherence was consistently observed among subgroups with either a high UPCI value (adjusted OR = 1.48, 95% CI 1.40 to 1.56) as well as those with a low UPCI value (adjusted OR = 1.60,

**Table 2. Measures of accuracy using UPCI[a] to predict USP[b], CMEP[c], and integrated COC[d] status.**

| | Sensitivity (95%CI[e]) | Specificity (95%CI[e]) | PPV[f] (95%CI[e]) | NPV[g] (95%CI[e]) | Kappa (95%CI[e]) |
|---|---|---|---|---|---|
| UPCI[a] to predict the usual statin prescriber | 0.55 (0.55, 0.56) | 0.61 (0.60, 0.62) | 0.78 (0.77, 0.78) | 0.35 (0.35, 0.36) | 0.13 (0.13, 0.14) |
| UPCI[a] to predict a CMEP[c] | 0.55 (0.54, 0.56) | 0.52 (0.51, 0.52) | 0.39 (0.39, 0.40) | 0.67 (0.66, 0.68) | 0.06 (0.05, 0.07) |
| UPCI[a] to predict integrated COC[d] | 0.58 (0.58, 0.59) | 0.53 (0.52, 0.53) | 0.33 (0.32, 0.33) | 0.76 (0.76, 0.77) | 0.09 (0.08, 0.09) |

[a] UPCI = usual provider continuity index;

[b] USP = usual statin prescriber;

[c] CMEP = complete medical examination provider;

[d] COC = continuity of care;

[e] CI = confidence interval;

[f] PPV = positive predictive value;

[g] NPV = negative predictive value.

95% CI 1.51 to 1.70). In contrast, the impact of a high UPCI appeared to have a weaker impact when tested in subgroups based on the presence or absence of integrated COC (OR = 1.13, 95% CI 1.06 to 1.21; and OR = 1.22, 95% CI 1.17 to 1.27; respectively) [Table 3]. Finally, patients receiving integrated COC with a low UPCI score had 31% higher odds of achieving optimal adherence versus those without integrated COC but a high UPCI value (OR = 1.31, 95%CI 1.24 to 1.39). In the Delong test, the adjusted model using the integrated COC term significantly improved the AUROC (+0.006, $\chi2$ statistic = 38.8, $p < 0.0001$) compared to the model using the UPCI measure of COC.

In sensitivity analyses, the effect of UPCI and integrated COC were similar to the primary analysis when the days supply of statin medications were not allowed to be accumulated between refills. Also, we changed the threshold to define "high UPCI" from the median value to the 25th percentile (cut-off = 0.62). In this case the association of the UPCI measure on optimal adherence was stronger (adjusted OR = 1.39, 95% CI 1.34 to 1.45) but weaker when the threshold was changed to the 75th percentile (cut-off = 1.00, adjusted OR = 1.09, 95% CI 1.05 to 1.13). Regardless of the threshold changes, the impact of the UPCI alone was still weaker than the effect by the integrated COC measure. The DeLong test showed that integrated COC term significantly improved the AUROC compared to the models using different thresholds of the UPCI measure (+0.004, $\chi2$ statistic = 16.0, $p < 0.0001$ when threshold was set at the 25th percentile of UPCI; +0.009, $\chi2$ statistic = 83.1, $p < 0.0001$ when threshold was set at the 75th percentile).

**Table 3. Odds ratios (OR[a]) and 95% confidence intervals (95% CI[b]) for the association of measures of COC[c] with optimal adherence (PDC[d] > = 80%).**

| | Unadjusted model OR (95%CI) | Adjusted model[g] ORa (95%CI) |
|---|---|---|
| Integrated COC[e] | 1.45 (1.40, 1.51) | 1.56 (1.50, 1.63) |
| Among patients with high UPCI[f] | | 1.48 (1.40, 1.56) |
| Among patients with low UPCI | | 1.60 (1.51, 1.70) |
| UPCI | 1.28 (1.24, 1.32) | 1.23 (1.19, 1.28) |
| Patients presenting integrated COC | | 1.13 (1.06, 1.21) |
| Patients not presenting integrated COC | | 1.22 (1.17, 1.27) |

[a]OR = odds ratio;

[b]CI = confidence interval;

[c]COC = continuity of care;

[d]PDC = proportion of days covered;

[e]Integrated COC = having a single physician identified as the usual care provider, the usual statin prescriber, and the complete medical examination provider;

[f]UPCI = usual provider continuity index;

[g]Covariates in the adjusted model were: 1) age, sex, residence (rural/urban), and income level (i.e., the neighborhood median household income quintile, lowest = 1, highest = 5) on the index date; 2) the following were measured within 365 days prior to the index date: number of hospitalizations, number of out-patient visits (to GPs and to specialists, respectively), number of emergency department visits, Charlson comorbidity score, number of distinct prescription medications (by drug identification numbers), and percentage of prescription medication cost paid by government health insurance; and 3) a list of chronic conditions identified between January 1st, 1996, and the index date, including osteoporosis, rheumatoid arthritis, hypertension, stroke, ischemic heart disease, acute myocardial infarction, heart failure, multiple sclerosis, Parkinson's disease, Alzheimer's disease and dementia, epilepsy, asthma, chronic obstructive pulmonary disease, diabetes, mood and anxiety diseases, schizophrenia, and cancer.

## Discussion

COC is considered to be an important determinant of medication adherence. It aligns with the paradigm of patient-centred care through coordination of services, especially when multiple providers are involved [15]. Quantitative studies appear to have confirmed this association with adherence; however, the most commonly used measure, the UPCI, is derived exclusively from the number of physician visits and fails to account for the coordination of care that is fundamental to the spirit of COC [6–9]. Our study was conducted exclusively on new statin users, yet the specific statin prescriber did not generate a high UPCI value in approximately half of the patients studied. However, adding statin prescribing and complete medical examination activities to the COC definition (i.e., with high UPCI value) resulted in a stronger association with medication adherence and significantly improved the predictive power of a medication adherence model. Further, the integrated COC measure added significant discrimination even when patients were stratified by high or low UPCI values. Our findings align with studies of patient-centered medical homes (PCMH) in which medication adherence appeared to be improved by care coordination [38, 43].

Despite the vast number of variables linked to medication adherence from published studies, almost all confer weak predictability in multivariable models [44–46]. Wong and colleagues conducted a population-based study with many patient-level factors including demographic characteristics (e.g., age, sex, and marital status), comorbidities (e.g., vascular disease, mental illness, and chronic lung or renal disorders), and regimen complexity [45]. The study also included clinical factors such as disease severity, and laboratory test results. Yet the authors found that all these variables only explained 2.9% of the adherence variation between patients [45]. Indeed, one of the major gaps in medication adherence research is the inability to explain more than a fraction of the variance observed with respect to adherence outcomes.

Population-based models of non-adherence generally have incomplete covariate adjustment and strategies are needed to capture clinical (e.g., side effects) and behavioural measures (e.g., attitudes and beliefs) [2]. However, our study demonstrated that improving on the measurement of existing variables may also help account for unexplained variance. The success of our integrated COC measure was likely due to the inclusion of diverse clinical services from a single practitioner rather than relying solely on the number of physician visits. However, it remains unknown whether the integrated COC measure could be improved further. Despite the identification of CMEP as a marker of COC in previous studies [17], it is possible that other service claim codes could replace (or be added to) CMEP. If complete medical examination claims were not necessary for optimal COC, then the contribution of this criterion was likely as a marker of diverse clinical service provision rather than a specific effect of CMEP itself. This issue is important to evaluate because the utilization and utility of CMEP may vary significantly depending on the jurisdiction/health system. We are hopeful our research will stimulate further work in the area and provide more information about the potential for health administrative databases to identify service patterns associated with primary care practices that promote optimal medication adherence.

Our study was not without limitations. First, the effect of COC on adherence was not adjusted by GP-related characteristics (e.g. age, sex, medical training background, workload, and prescribing habits), although the literature suggests that these characteristics may affect medication adherence [16–18, 47]. Second, GP physicians paid on salary (rather than fee-for service) are not required to submit service claims. As a result, the number of visits may have been underestimated. Alternatively, GPs paid by a fixed salary may perform differently than GPs paid by fee-for-service but that factor could not be assessed [48]. Third, the association between the integrated COC measures and adherence may not be causal. Having all services

from the same GP could be a sign of a successful relationship rather than its cause. Nonetheless, COC measures have been used in many studies on medication adherence with positive findings and have a strong theoretical link to the origins of the problem [6–9].

Despite these limitations, we improved upon an existing measure of COC that not only produced a robust odds ratio, but also improved the predictive success of an adherence model containing a large number of established covariates. Thus, our findings do not merely identify a new variable for models of medication adherence but they contribute to an important and elusive goal of explaining the phenomenon within a framework that remains highly theoretical.

## Conclusion

The most common approach for measuring COC in adherence models fails to account for a key principle of service coordination. An updated measure that requires evidence for other clinical services (i.e., prescribing and medical examinations) with physician visits is more consistent with the concept of COC and the value of patient-centred care. In addition, the use of an integrated measure of COC provided better discrimination of adherent patients and improved predictive performance of a covariate-adjusted adherence model. An integrated measure should be considered as the standard approach for representing COC in population-based adherence models.

## Acknowledgments

The authors acknowledge Saskatchewan Health Quality Council for use of de-identified data provided by the Saskatchewan Ministry of Health and eHealth Saskatchewan. The interpretation and conclusions contained herein do not necessarily represent those of the Government of Saskatchewan, the Saskatchewan Ministry of Health, or eHealth Saskatchewan.

## Author Contributions

**Conceptualization:** Shenzhen Yao, Lisa Lix, Gary Teare, Charity Evans, David Blackburn.

**Data curation:** Shenzhen Yao.

**Formal analysis:** Shenzhen Yao, Lisa Lix.

**Funding acquisition:** David Blackburn.

**Investigation:** Shenzhen Yao, David Blackburn.

**Methodology:** Shenzhen Yao, Lisa Lix, Gary Teare, Charity Evans, David Blackburn.

**Project administration:** Shenzhen Yao, David Blackburn.

**Resources:** Shenzhen Yao, David Blackburn.

**Software:** Shenzhen Yao.

**Supervision:** Lisa Lix, Gary Teare, Charity Evans, David Blackburn.

**Validation:** Shenzhen Yao, Lisa Lix, Gary Teare, Charity Evans, David Blackburn.

**Visualization:** Shenzhen Yao, David Blackburn.

**Writing – original draft:** Shenzhen Yao, David Blackburn.

**Writing – review & editing:** Shenzhen Yao, Lisa Lix, Gary Teare, Charity Evans, David Blackburn.

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
