## [Decision Letter · Decision Letter 0]

7 Jun 2021

PONE-D-21-07353

The impact of continuity of care on medication adherence: a population based study

PLOS ONE

Dear Dr. Blackburn,

Thank you for submitting your manuscript to PLOS ONE. After careful consideration, we feel that it has merit but does not fully meet PLOS ONE’s publication criteria as it currently stands. Therefore, we invite you to submit a revised version of the manuscript that addresses the points raised during the review process.

We look forward to receiving your revised manuscript.

Kind regards,

George Liu, PhD

Academic Editor

PLOS ONE

Journal Requirements:

2. Please provide additional details regarding participant consent. In the ethics statement in the Methods and online submission information, please ensure that you have specified (i) whether consent was informed and (ii) what type you obtained (for instance, written or verbal, and if verbal, how it was documented and witnessed). If your study included minors, state whether you obtained consent from parents or guardians. If the need for consent was waived by the ethics committee, please include this information.

'I have read the journal's policy and the authors of this manuscript have the following competing interests: David Blackburn is the Chair in Patient Adherence to Drug Therapy within the College of Pharmacy and Nutrition, University of Saskatchewan.  This position was created through unrestricted financial support from AstraZeneca Canada, Merck Canada, Pfizer Canada, and the Province of Saskatchewan’s Ministry of Health.  None of the sponsors were involved in developing this study or writing the manuscript. Shenzhen Yao, Lisa Lix, Gary Teare, and Charity Evans declare no conflicts.  '

5. Please ensure that you refer to Figure 1 in your text as, if accepted, production will need this reference to link the reader to the figure.

6. Please include a copy of Table 2 which you refer to in your text on page 14.

Reviewers' comments:

Reviewer's Responses to Questions

**Comments to the Author**

1. Is the manuscript technically sound, and do the data support the conclusions?

Reviewer #1: Yes

Reviewer #2: Yes

2. Has the statistical analysis been performed appropriately and rigorously? 

Reviewer #1: Yes

Reviewer #2: Yes

3. Have the authors made all data underlying the findings in their manuscript fully available?

Reviewer #1: Yes

Reviewer #2: No

4. Is the manuscript presented in an intelligible fashion and written in standard English?

Reviewer #1: Yes

Reviewer #2: Yes

5. Review Comments to the Author

Reviewer #1: Thanks very much for this opportunity to review this interesting study. This study aimed to improve the measure of COC, based on a more comprehensive perspective, and use it to predict medication adherence. The results comparing the original measures of COC and this new-developed measure also suggested its advantage on predicting medication adherence and give some implications to clinical practices. Here are some concerns raised for authors to consider.

Background

1. The introduction of the current study seemed insufficient, especially some basic background information missed. Considering the aim of the current study is to improve the medication adherence, the prevalence of non-adherence and its consequences should be added to highlight the significance of the current study. In addition, medication adherence is a complex process and authors may need to briefly summary it and further swift the focus on the effect of COC.

2. The introduction mentioning the mediation effect of patient-physician relationship (1st paragraph) seemed useless based on the aim and design of the current study. Why authors have to mentioned this effect if this kind of factors were not considered/adjusted in the current study?

Methods

3. (Data sources) It seemed the median household income quintiles is estimated by linking the first three digits of postal code to Statistics Canada Census data. Pls provide a further interpretation for the validity of this method and relevant references.

4. (Study design and population) The cohort period seemed from 2012-2017, while this study tried to identify new initiation as receiving no dispensations for a statin medication in the five years prior. I noticed that the data ranges from 1999. It seemed confused why authors choose this period for construction of the current cohort rather than the overall period of data available (for example, 2004-2017 to ensure 5 years prior dispensing data). Is there any specific reason?

5. (Study design and population) The reasons why statin medications were selected in the current study may be insufficient. Though some aspects were mentioned by authors, however, why these aspects may influence the selection of medication should be further interpreted (as well as references).

6. The number of exclusion criteria is disordered, pls re-number them.

Results

7. (Table 3) mis-writing of “Among patients with high UPCIf”

Discussion

8. Considering the main contribution of the current study focused on the newly-developed measure of COC and its application. It would be necessary for authors to further discuss its implication for clinics and future research. Is the comprehensive measure possible for further use to improve COC, what is the advantage as well as potential unintended consequences of this COC measures compared with existing UPCI and etc..

Reviewer #2: Overall, this paper is clear and presents a well-executed study on continuity of care (COC) measures and their relationship to medication adherence. The case is well-made for the value of a COC measure that integrates usual statin provider and complete medical examination provider with the conventional usual care provider concept.

I have several specific suggestions for improving the manuscript:

1. More specific title. The paper is really about the value of an integrated COC measure. I think that should be in the title.

2. It’d be good if it were easier for the reader to see that ‘integrated COC measure’ means integrating usual statin and complete medical examination provider with usual GP – ideally in the abstract, and I think also in the conclusion. The phrasing from the last paragraph of the Introduction, “integrated COC measure consisting of physician visits, prescribing, and claims for a complete medical examination” is a good example of language that clarifies what the integrated COC measure is, but I feel it should be more prominent given that it’s essentially the subject of the paper.

3. Results Fig 1 – cohort flowchart is missing in the manuscript I reviewed.

4. Evidently a prescriber ID is in the dispense record (prescription drug claims file), but this is not stated – see last sentence of Data Sources.

5. The word ‘included’ is used repeatedly in Covariates (and also in footnote g of table 3) – are these all the covariates or just examples from a much longer list? If the list is complete please say ‘were’ (e.g. ‘The patient covariates were age, sex, and residence (rural/urban) on the index date’). Perhaps include an appendix or supplementary material if the covariates in the adjusted model are too extensive to list in the body of the paper.

6. Results: The authors say 60.1% had no complete medical exam or received it from a specialist – can we have a breakdown between those alternatives?

7. Not enough is made of the sensitivity of predictor performance to the threshold used in defining ‘high UPCI’. The median for UPCI is reported as a high-sounding 0.82, whereas integrated COC is rarer at 28.3% - is this why it’s a better predictor of adherence? Apparently not entirely… it’s great that you’ve included the better performance with 25th percentile UPCI – OR 1.39 v 1.56 instead of 1.23 v 1.56. (Incidentally this seems consistent with lowest 33% of UPCI being distinct from middle and highest 33% for adherence in your reference [4].) That said, you didn’t go back to see how this more effective UPCI cut-off performed for predicting enhanced COC, or usual statin or complete medical exam provider.

8. As a more general comment on the point above, the use of data from one particular healthcare system should be cited as a limitation. You have shown that it’s possible to formulate an effective integrated COC measure quite elegantly in the Saskatchewan health system. But this measure is surely sensitive to the rates at which complete medical examinations are performed as well as the median (and distribution) of UPCI. I’m not an authority on this, but I take it that comprehensive annual medical checks are normal in, say, Japan; and I presume they’re much rarer in some other jurisdictions. Similarly, the role of the GP is strong in much of the commonwealth, but may not extend to regular statin provision in other systems. At any rate, please note the use of data from one system as a limitation, and consider further discussion of how the model may or may not apply more broadly.

9. Typo: Last paragraph of Discussion, ‘we improved up on’ -> ‘we improved upon’

6. PLOS authors have the option to publish the peer review history of their article (what does this mean?). If published, this will include your full peer review and any attached files.

Reviewer #1: **Yes: **Chenxi Liu

Reviewer #2: No

---

## [Author Response · Author response to Decision Letter 0]

8 Dec 2021

I can confirm that my competing interests statement does not alter our adherence to PLOS ONE policies on sharing data and materials

I have included this statement in the cover letter also. If you require any further clarification please don't hesitate to contact me.

---

## [Decision Letter · Decision Letter 1]

17 Jan 2022

PONE-D-21-07353R1An integrated continuity of care measure improves performance in models predicting medication adherence using population-based administrative dataPLOS ONE

Dear Dr. Blackburn,

Thank you for submitting your manuscript to PLOS ONE. After careful consideration, we feel that it has merit but does not fully meet PLOS ONE’s publication criteria as it currently stands. Therefore, we invite you to submit a revised version of the manuscript that addresses the points raised during the review process.

Please address the additional comments made by reviewer #1

We look forward to receiving your revised manuscript.

Kind regards,

George Liu, PhD

Academic Editor

PLOS ONE

Reviewers' comments:

Reviewer's Responses to Questions

**Comments to the Author**

1. If the authors have adequately addressed your comments raised in a previous round of review and you feel that this manuscript is now acceptable for publication, you may indicate that here to bypass the “Comments to the Author” section, enter your conflict of interest statement in the “Confidential to Editor” section, and submit your "Accept" recommendation.

Reviewer #1: All comments have been addressed

Reviewer #2: All comments have been addressed

2. Is the manuscript technically sound, and do the data support the conclusions?

Reviewer #1: Yes

Reviewer #2: Yes

3. Has the statistical analysis been performed appropriately and rigorously? 

Reviewer #1: Yes

Reviewer #2: Yes

4. Have the authors made all data underlying the findings in their manuscript fully available?

Reviewer #1: Yes

Reviewer #2: Yes

5. Is the manuscript presented in an intelligible fashion and written in standard English?

Reviewer #1: Yes

Reviewer #2: Yes

6. Review Comments to the Author

Reviewer #1: Thanks very much for this opportunity to review the revised draft of the current study. It’s much clear this time. There were only a few minor concerns for authors as follows.

1. The presentation of data sources could be improved. Authors may use subsections to make the presentation clearer (based on 5 files) or add a table to summarize this information.

2. Discussion, 1st paragraph, “Our results indicate that a high UPCI value did not identify physicians who were providing core services”. It seemed that the statement is not accurate. First, the current study showed that the UPCI index was partly able to predict other indicators (table 2); Second, it seemed that defining prescribing and examination as core services is too arbitrary. Thus, the statement may be revised to accurately reflect the improvement of integrated COC compared with UPCI.

Reviewer #2: The revisions are entirely to my satisfaction in addressing prior concerns. I'm particularly happy with the changes to the title, abstract and discussion.

One very minor point with the new sensitivity analysis text: "The DeLong test showed that integrated COC term

significantly improved the AUROC compared to the models using different thresholds of the UPCI measure (+0.004, χ2 statistic = 16.0, p < 0.0001 when threshold was set at the 25th percentile of UPCI; +0.009, χ2 statistic = 16.0, p < 0.0001 when threshold was set at the 75th percentile)" I'm suspicious of both chi-squareds being 16.0 given the different changes in AUROC. Perhaps the 25th percentile one is 16.0 and the 75th percentile one is somewhat higher?? Just pointing out for the authors to check.

7. PLOS authors have the option to publish the peer review history of their article (what does this mean?). If published, this will include your full peer review and any attached files.

Reviewer #1: No

Reviewer #2: **Yes: **James R Warren

---

## [Author Response · Author response to Decision Letter 1]

21 Jan 2022

Itemized response to reviewers has been uploaded. Please let us know if further clarification is required

---

## [Decision Letter · Decision Letter 2]

7 Feb 2022

An integrated continuity of care measure improves performance in models predicting medication adherence using population-based administrative data

PONE-D-21-07353R2

Dear Dr. Blackburn,

We’re pleased to inform you that your manuscript has been judged scientifically suitable for publication and will be formally accepted for publication once it meets all outstanding technical requirements.

Kind regards,

Juan F. Orueta, MD, PhD

Academic Editor

PLOS ONE

Additional Editor Comments:

There is a typo on page 17, line 8. The word "versus" is repeated and the parenthesis sign is misplaced.

Reviewers' comments:

Reviewer's Responses to Questions

**Comments to the Author**

1. If the authors have adequately addressed your comments raised in a previous round of review and you feel that this manuscript is now acceptable for publication, you may indicate that here to bypass the “Comments to the Author” section, enter your conflict of interest statement in the “Confidential to Editor” section, and submit your "Accept" recommendation.

Reviewer #2: All comments have been addressed

2. Is the manuscript technically sound, and do the data support the conclusions?

Reviewer #2: Yes

3. Has the statistical analysis been performed appropriately and rigorously? 

Reviewer #2: Yes

4. Have the authors made all data underlying the findings in their manuscript fully available?

Reviewer #2: Yes

5. Is the manuscript presented in an intelligible fashion and written in standard English?

Reviewer #2: Yes

6. Review Comments to the Author

Reviewer #2: (No Response)

7. PLOS authors have the option to publish the peer review history of their article (what does this mean?). If published, this will include your full peer review and any attached files.

Reviewer #2: No

---

## [Editor Report · Acceptance letter]

16 Feb 2022

PONE-D-21-07353R2 

An integrated continuity of care measure improves performance in models predicting medication adherence using population-based administrative data 

Dear Dr. Blackburn:

I'm pleased to inform you that your manuscript has been deemed suitable for publication in PLOS ONE. Congratulations! Your manuscript is now with our production department. 

Kind regards, 

on behalf of

Dr. Juan F. Orueta 

Academic Editor

PLOS ONE